# BAYESIAN BINARY SEARCH

## ABSTRACT

We present Bayesian Binary Search (BBS), a novel probabilistic variant of the classical binary search/bisection algorithm. BBS leverages machine learning/statistical techniques to estimate the probability density of the search space and modifies the bisection step to split based on probability density rather than the traditional midpoint, allowing for the learned distribution of the search space to guide the search algorithm. Search space density estimation can flexibly be performed using supervised probabilistic machine learning techniques (e.g., Gaussian process regression, Bayesian neural networks, quantile regression) or unsupervised learning algorithms (e.g., Gaussian mixture models, kernel density estimation (KDE), maximum likelihood estimation (MLE)). We demonstrate significant efficiency gains of using BBS on both simulated data across a variety of distributions and in a real-world binary search use case of probing channel balances in the Bitcoin Lightning Network, for which we have deployed the BBS algorithm in a production setting.

## 1 INTRODUCTION

The concept of organizing data for efficient searching has ancient roots. One of the earliest known examples is the Inakibit-Anu tablet from Babylon (c. 200 BCE), which contained approximately 500 sorted sexagesimal numbers and their reciprocals, facilitating easier searches Knuth (1998). Similar sorting techniques were evident in name lists discovered on the Aegean Islands. The Catholicon, a Latin dictionary completed in 1286 CE, marked a significant advance by introducing rules for complete alphabetical classification Knuth (1998).

The documented modern era of search algorithms began in 1946 when John Mauchly first mentioned binary search during the seminal Moore School Lectures Knuth (1998). This was followed by William Wesley Peterson's introduction of interpolation search in 1957 Peterson (1957). A limitation of early binary search algorithms was their restriction to arrays with lengths one less than a power of two. This constraint was overcome in 1960 by Derrick Henry Lehmer, who published a generalized binary search algorithm applicable to arrays of any length Lehmer (1960). A significant optimization came in 1962 when Hermann Bottenbruch presented an ALGOL 60 implementation of binary search that moved the equality comparison to the end of the process Bottenbruch (1962). While this increased the average number of iterations by one, it reduced the number of comparisons per iteration to one, potentially improving efficiency. Further refinements to binary search continued, with A. K. Chandra of Stanford University developing the uniform binary search in 1971 Knuth (1998). This variant aimed to provide more consistent performance across different input distributions. In the realm of computational geometry, Bernard Chazelle and Leonidas J. Guibas introduced fractional cascading in 1986 Chazelle & Guibas (1986). This technique provided a method to solve various search problems efficiently in geometric contexts, broadening the application of search algorithms beyond simple sorted lists. More recently, (Mohammed et al., 2021), proposed a hybrid search algorithm combining binary search and interpolated search to optimize search on datasets with non-uniform distributions.

The bisection method, a fundamental technique in numerical analysis, shares conceptual similarities with binary search. This method, also known as the interval halving method, binary chopping, or Bolzano's method, has origins of the bisection method can be traced back to ancient mathematics. However, its formal description is often attributed to Bolzano's work in the early 19th century Russ (2004). Bolzano's theorem, which guarantees the existence of a root in a continuous function that changes sign over an interval, provides the theoretical basis for the bisection method. In the context of root-finding algorithms, Burden & Faires (2015) provide a comprehensive treatment of the bisec-

tion method, discussing its convergence properties and error bounds. They highlight the method's robustness and guaranteed convergence, albeit at a relatively slow linear rate. Variants of the bisection method have been developed to improve efficiency and applicability. Horstein (1963) first introduced probabilistic bisection. Hansen & Walster (1991) employed interval analysis and bisection to develop reliable methods for finding global optima. Their work demonstrates how bisection principles can be extended beyond simple root-finding to more complex optimization problems.

The connection between bisection and binary search in computer science was explored by Knuth (1998), who discussed both methods in the context of searching and optimization algorithms. This connection underlies our approach in leveraging probabilistic techniques to enhance binary search. Recent work has focused on adapting bisection methods to handle uncertainty and noise. Waeber et al. (2013) introduced a probabilistic bisection algorithm for noisy root-finding problems. Their method, which updates a probability distribution over the root location based on noisy measurements, shares conceptual similarities with our Bayesian Binary Search approach. In the context of stochastic optimization, Jedynak et al. (2012) developed a probabilistic bisection algorithm for binary classification problems. Their work demonstrates how Bayesian updating can be incorporated into the bisection process, providing a precedent for our probabilistic approach to binary search. The bisection method has also found applications in various fields beyond pure mathematics. For instance, Nievergelt (1964) discussed the use of parallel bisection methods in scientific computing, highlighting the potential for algorithmic improvements through parallelization.

Probabilistic machine learning, as described by Ghahramani (2015), provides a framework for reasoning about uncertainty in data and models. Gaussian Processes (GPs), a cornerstone of many probabilistic machine learning methods, were extensively explored by Rasmussen & Williams (2006). GPs offer a flexible, non-parametric approach to modeling distributions over functions, making them particularly suitable for tasks involving uncertainty quantification. In recent years, Bayesian deep learning has emerged as a powerful approach to combining the expressiveness of deep neural networks with principled uncertainty quantification. Wang & Yeung (2016) provide a comprehensive survey of Bayesian deep learning techniques, discussing various methods for incorporating uncertainty into deep neural networks and their applications in complex, high-dimensional problems. The field of Bayesian Optimization, which shares conceptual similarities with our work, has seen substantial growth. Shahriari et al. (2015) provide a comprehensive overview of Bayesian Optimization techniques, highlighting their effectiveness in optimizing expensive-to-evaluate functions. These methods typically use GPs, Bayesian Neural Networks (BNNs) or other probabilistic models to guide the optimization process, analogous to our use of probability density estimates in Bayesian Binary Search.

## 2 METHODOLOGY

### 2.1 CLASSICAL BINARY SEARCH/BISECTION

Binary search is an efficient algorithm for finding a target value in a defined search space. It repeatedly divides the search interval in half, eliminating half of the remaining space at each step. The algorithm maintains two boundaries, low and high, and computes the midpoint mid = (low + high) / 2. By evaluating the midpoint and comparing it to the target value, it determines which half of the search space to explore next, updating either low or high accordingly. This process continues until the target is found or the search space is exhausted. The efficiency of binary search stems from its ability to halve the potential search space with each iteration, resulting in a logarithmic time complexity.

### 2.2 BAYESIAN BINARY SEARCH (BBS)

#### 2.2.1 PROBLEM FORMULATION

Given a search space $S$ and a target value $t$, our aim is to locate $t$ in $S$, or produce a specified range for t with a given tolerance. Unlike classical binary search, we assume that the distribution of potential target locations is not uniform across $S$. We represent this non-uniform distribution using a probability density function (PDF) $p(x)$, where $x \in S$. We formulate the binary search algorithm in two parts: search space density estimation, which produces a PDF that is fed into a modified binary search algorithm which begins at the median of the PDF and bisects in probability density space. BBS is equivalent to basic binary search when the search space PDF estimation process is replaced with the assumption of a uniformly distributed target space. This formulation maps to a binary search problem on a sorted array in the following way: the search space S corresponds to the

sorted array, the target value t is the element being searched for, and the PDF p(x) represents the probability of finding the target at each index in the array using interpolation.

## 2.3 SEARCH SPACE PROBABILITY DENSITY FUNCTION ESTIMATION

The effectiveness of BBS depends on the accuracy of the PDF estimation. Some methods for estimating $p(x)$ can include:

### 2.3.1 SUPERVISED LEARNING APPROACHES

- **Gaussian Process Regression (GPR):** GPR provides a non-parametric way to model the PDF, offering both a mean prediction and uncertainty estimates.

- **Bayesian Neural Networks (BNN):** BNNs combine the flexibility of neural networks with Bayesian inference, allowing for uncertainty quantification in the predictions.

- **Quantile Regression:** Quantile regression estimates the conditional quantiles of a response variable, providing a more comprehensive view of the relationship between variables across different parts of the distribution, without assuming a particular parametric form for the underlying distribution.

### 2.3.2 UNSUPERVISED LEARNING APPROACHES

- **Gaussian Mixture Models (GMM):** GMMs (typically fit using the Expectation-Maximization (EM) algorithm) can model complex, multimodal distributions by representing the PDF as a weighted sum of Gaussian components.

- **Kernel Density Estimation (KDE):** KDE is a non-parametric method for estimating PDFs, which can capture complex shapes without assuming a specific functional form.

- **Maximum Likelihood Estimation (MLE):** MLE is a method of estimating the parameters of a probability distribution by maximizing a likelihood function. It can be used to fit various parametric distributions (e.g., normal, exponential, Poisson) to data, providing a PDF that best explains the observed samples according to the chosen distribution family.

## 2.4 BINARY SEARCH WITH PROBABILISTIC BISECTION

BBS modifies the classical binary search by replacing the midpoint calculation with a probabilistic bisection step. Instead of dividing the search space at the midpoint, BBS divides it at the median of the current PDF. Formally, at each step, we find $x^*$ such that:

$$\int_{low}^{x^*} p(x)dx = \int_{x^*}^{high} p(x)dx = 0.5 \tag{1}$$

After each comparison, we update the PDF based on the result. If the target is found to be in the lower half, we set $p(x) = 0$ for $x > x^*$ and renormalize the PDF. Similarly, if the target is in the upper half, we set $p(x) = 0$ for $x < x^*$ and renormalize.

---

**Algorithm 1:** Bayesian Binary Search

---

**Input:** low $\in \mathbb{Z}$, high $\in \mathbb{Z}$, $\epsilon > 0$
**Output:** Target bound with $\epsilon$ tolerance, or -1 if not found

1 $p(x) \leftarrow \texttt{EstimatePDF}(low, high)$
2 **while** *low $\leq$ high* **do**
3    $x^* \leftarrow \texttt{FindMedian}(p(x), low, high)$
4    $s \leftarrow \texttt{Sign}(x^*)$
5    **if** *high $-$ low $\leq \epsilon$* **then**
6      **return** *low, high*
7    **else if** *s $> 0$* **then**
8      *high $\leftarrow x^*$*
9    **else**
10      *low $\leftarrow x^*$*
11    **end**
12    $p(x) \leftarrow \texttt{UpdatePDF}(p(x), low, high)$
13 **end**
14 **return** $-1$;
15

---

**Algorithm 2:** Estimate Search Space Probability Density Function (PDF)

---

1 **Function** `EstimatePDF` (*low, high*):
```
    // Estimate the initial PDF based on the search space
    // This implementation can vary depending on the available
       data but can include supervised probabilistic machine
       learning algorithms or unsupervised statistical methods
       for distribution estimation
```
2    **return** $p(x)$

---

**Algorithm 3:** FindMedian Function

---

1 **Function** `FindMedian` (*p(x), low, high*):
2    **return** $\arg\min_{x \in [\text{low,high}]} \left| \int_{\text{low}}^{x} p(t)dt - 0.5 \right|$

---

**Algorithm 4:** UpdatePDF Function

---

1 **Function** `UpdatePDF` (*p(x), low, high*):
2    **if** $x \in [low, high]$ **then**
3      $p_{\text{new}}(x) \leftarrow \frac{p(x)}{\int_{\text{low}}^{\text{high}} p(t)\, dt}$
4    **else**
5      $p_{\text{new}}(x) \leftarrow 0$
6    **end**
7    **return** $p_{\text{new}}(x)$

---

**Algorithm 5:** Sign Function

---

1 **Function** `Sign` (*x*):
```
    // This function should be implemented based on the specific
       problem
    // It should return -1 if x is less than or equal to the
       target, 1 if x is greater than the target
```
2    **if** *x $>$ target* **then**
3      **return** 1
4    **else**
5      **return** -1
6    **end**

We evaluate BBS against basic binary search on both simulated data across several distributions and a real world example of binary search/bisection of probing channels in the Bitcoin Lightning Network, for which accessing the search space is an expensive operations (time and liquidity) Tikhomirov et al. (2020).

## 2.5 EXPERIMENTS ON SIMULATED DATA

We evaluate the performance of BBS against binary search on simulated data from several distributions: normal, bimodal and exponential. The results of the normal distribution experiments are shown below, while the results of the bimodal and exponential distributions can be found in the appendix. We additionally show experiments for which the estimated search space density function does not match the target search space as measured by Kullback-Leibler (KL) Divergence. We demonstrate how BBS performance degrades as the estimated search space density drifts further from the target distribution, and eventually can cause the BBS to perform worse than basic binary search.

### 2.5.1 EXPERIMENTAL SETUP

To evaluate the performance of BBS compared to the standard binary search in the context of normal distributions, we conducted a series of experiments with the following setup:

### 2.5.2 DISTRIBUTION PARAMETERS

For the normal distribution experiments, we used the following parameters:

- **Mean** ($\mu$): 0

- **Standard Deviation** ($\sigma$): 10000

### 2.5.3 TARGET VALUES GENERATION

We generated 1,000 target values by sampling from the specified normal distribution. Each target value was obtained by drawing a random sample $x$ from the distribution and applying the floor function to ensure integer targets: $target = \text{floor}(x)$.

### 2.5.4 SEARCH PROCEDURE

For each target value, both search algorithms attempt to locate the target within a specified precision ($\epsilon$), set to 8. The search space is initialized based on the properties of the normal distribution:

- **Lower Bound** (*lo*): $\text{floor}(\mu - 4.2 \times \sigma)$

- **Upper Bound** (*hi*): $\text{ceil}(\mu + 4.2 \times \sigma)$

The algorithms iteratively narrow down the search space until the target is found within the acceptable error margin.

### 2.5.5 METRICS COLLECTED

We collected the following performance metrics:

- **Number of Steps**: Total iterations required to find each target.

- **Bracket Size**: The range ($hi - lo$) at each step of the search.

Table 1: $\mu = 0.0, \sigma = 10000.0, N = 500$

| $\epsilon$ | Percent Decrease | Basic Mean Steps | BBS Mean Steps |
|---|---|---|---|
| 1 | 6.30% | $16.47 \pm 0.50$ | $15.43 \pm 1.19$ |
| 2 | 6.36% | $15.68 \pm 0.47$ | $14.68 \pm 1.15$ |
| 3 | 6.17% | $15.00 \pm 0.00$ | $14.07 \pm 1.15$ |
| 4 | 8.76% | $15.00 \pm 0.00$ | $13.69 \pm 1.11$ |
| 5 | 5.06% | $14.15 \pm 0.36$ | $13.43 \pm 1.02$ |
| 6 | 6.16% | $14.00 \pm 0.00$ | $13.14 \pm 1.09$ |
| 7 | 8.07% | $14.00 \pm 0.00$ | $12.87 \pm 1.16$ |
| 8 | 9.39% | $14.00 \pm 0.00$ | $12.69 \pm 1.12$ |
| 9 | 10.27% | $14.00 \pm 0.00$ | $12.56 \pm 1.08$ |
| 10 | 6.14% | $13.28 \pm 0.45$ | $12.47 \pm 1.01$ |
| 11 | 4.63% | $13.00 \pm 0.00$ | $12.40 \pm 0.99$ |
| 12 | 5.88% | $13.00 \pm 0.00$ | $12.24 \pm 1.05$ |
| 13 | 7.43% | $13.00 \pm 0.00$ | $12.03 \pm 1.13$ |
| 14 | 8.51% | $13.00 \pm 0.00$ | $11.89 \pm 1.15$ |
| 15 | 9.32% | $13.00 \pm 0.00$ | $11.79 \pm 1.15$ |
| 16 | 9.89% | $13.00 \pm 0.00$ | $11.71 \pm 1.12$ |
| 17 | 10.49% | $13.00 \pm 0.00$ | $11.64 \pm 1.11$ |
| 18 | 11.03% | $13.00 \pm 0.00$ | $11.57 \pm 1.09$ |
| 19 | 11.42% | $13.00 \pm 0.00$ | $11.52 \pm 1.05$ |
| 20 | 8.38% | $12.53 \pm 0.50$ | $11.48 \pm 1.03$ |
| 21 | 4.52% | $12.00 \pm 0.00$ | $11.46 \pm 1.01$ |
| 22 | 4.80% | $12.00 \pm 0.00$ | $11.42 \pm 0.97$ |
| 23 | 5.07% | $12.00 \pm 0.00$ | $11.39 \pm 0.95$ |
| 24 | 5.83% | $12.00 \pm 0.00$ | $11.30 \pm 1.01$ |
| 25 | 7.12% | $12.00 \pm 0.00$ | $11.15 \pm 1.10$ |
| 26 | 8.03% | $12.00 \pm 0.00$ | $11.04 \pm 1.13$ |
| 27 | 8.43% | $12.00 \pm 0.00$ | $10.99 \pm 1.14$ |
| 28 | 9.17% | $12.00 \pm 0.00$ | $10.90 \pm 1.15$ |
| 29 | 9.55% | $12.00 \pm 0.00$ | $10.85 \pm 1.15$ |
| 30 | 10.02% | $12.00 \pm 0.00$ | $10.80 \pm 1.15$ |
| 31 | 10.33% | $12.00 \pm 0.00$ | $10.76 \pm 1.14$ |
| 32 | 10.57% | $12.00 \pm 0.00$ | $10.73 \pm 1.13$ |

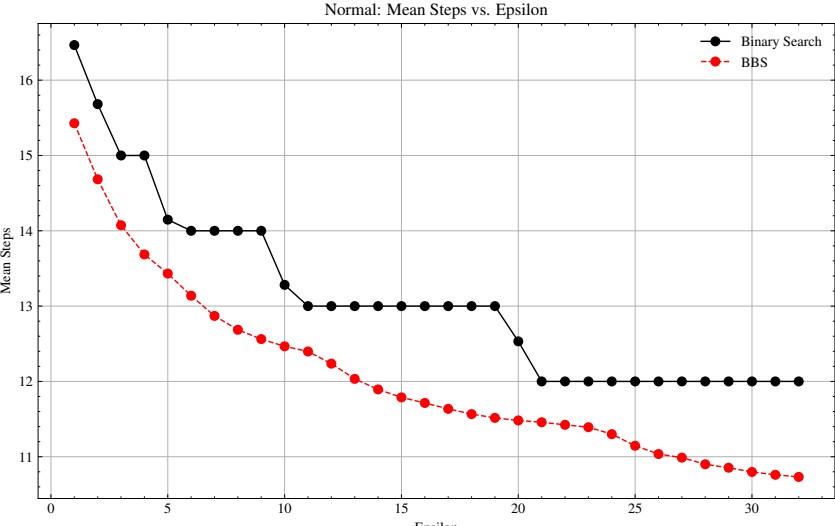

Figure 1: Normal Distribution: Basic vs. BBS Convergence Comparison

## 2.6 EXPERIMENTS ON LIGHTNING NETWORK CHANNEL PROBING

Probing a channel in the Lightning Network is the construction of a fake payment to be sent through the network to obtain information on the balance of a channel. Typically, it is currently performed using a basic binary search/bisection. The response of a given probe can be viewed as an oracle, which is a continuous and monotonic function, making it suitable for the bisection method as it guarantees there is only one root of the function.

Each probe is computationally expensive and fundamentally constrained in this domain due to the max HTLC (Hashed Timelock Contract) limit (483 on Lightning Network Daemon (LND), the most popular Lightning Node implementation) on each channel in the Lightning node software implementation for security purposes. Probes occupying this limit fundamentally constrain network throughput, which could otherwise process real payments.

To demonstrate the computational complexity of executing probes on the Lightning Network, we probed 1500 channels from our Lightning node, with an average hop length of 2.26. The probe time average for each channel had a mean of 3.1 seconds and a standard deviation of 0.7 seconds. The density estimation step per channel here in comparison (random forest inference and KDE) takes 0.18 seconds, and only needs to be done once per channel. The added overhead of density estimation here is significantly outweighed by the probing computational cost and the domain max HTLC constraint.

For search space density estimation in the Lightning Network channel probing experiment, we detail a random forest model to predict the search target (channel balance in this case). We construct a PDF from the prediction of the random forest using kernel density estimation (KDE) on the predictions of individual predictors (trees) in the ensemble. We alternatively could use Bayesian Neural Networks or a Gaussian Process Regressor (results in appendix), but the random forest yielded superior predictive performance in our experiments (Rossi et al., 2024).

### 2.6.1 DESCRIPTION OF CHANNEL BALANCE PREDICTION TASK

The Lightning Network can be modeled as a directed graph $G = (V, E)$, where $V$ represents the set of nodes and $E$ represents the set of edges. Each node $u$ and each edge $(u, v)$ have associated features, denoted by $\mathbf{x}_u \in \mathbb{R}^k$ for nodes and $\mathbf{e}_{(u,v)}$ for edges, which contain specific information about those nodes and edges. Additionally, each edge $(u, v)$ has a scalar value $y_{(u,v)} \geq 0$, representing the pre-allocated balance for transactions from $u$ to $v$.

Graph $G$ has the constraint that if an edge exists in one direction, it must also exist in the opposite direction, i.e., $(u, v) \in E \Leftrightarrow (v, u) \in E$. The set of two edges between any two nodes is called a channel, denoted as $\{(u, v), (v, u)\}$. For simplicity, we represent a channel by the set of its two nodes: $\{u, v\}$. The total capacity of the channel $\{u, v\}$ is defined as $c_{\{u,v\}} = y_{(u,v)} + y_{(v,u)}$.

We are provided with the total channel capacities $c_{\{u,v\}}$ for all channels in the graph, but we only know the individual values $y_{(u,v)}$ for a subset of edges. Note that knowing $y_{(u,v)}$ allows us to determine $y_{(v,u)}$, since $y_{(v,u)} = c_{\{u,v\}} - y_{(u,v)}$. Therefore, we can focus on predicting $y_{(u,v)}$.

Moreover, since we are given $c_{\{u,v\}}$ for all edges, and we know $0 \leq y_{(u,v)} \leq c_{\{u,v\}}$, we also have that $y_{(u,v)} = p_{(u,v)} c_{\{u,v\}}$, where $0 \leq p_{(u,v)} \leq 1$. Intuitively, $p_{(u,v)}$ is the proportion of the channel capacity which belongs to the $(u, v)$ direction. From this, we see that we focus on predicting $p_{(u,v)}$, and then use it to obtain $y_{(u,v)}$.

Therefore, our primary task is to predict $p_{(u,v)}$ for all edges where it is not observed.

### 2.6.2 DATA COLLECTION AND PREPROCESSING

The data used in this experiment is a combination of publicly available information from the Lightning Network and crowdsourced information from nodes in the network. A snapshot of the network from December 15th, 2023 is used in this experiment. Balance information for each node is represented by its local balance, recorded at one-minute intervals over the preceding hour. This data is converted into a probability density function (PDF) through kernel density estimation. Subsequently, a balance value is sampled from each PDF, serving as the representative local balance for the corresponding channel within the dataset.

### 2.7 METHODOLOGY

The following section outlines the details of our modeling approach.

### 2.7.1 MODELING

We will predict $p_{(u,v)}$ by learning a parametric function:

$$\hat{p}_{(u,v)} = f_\Theta(u, v, G, \mathbf{x}_u, \mathbf{x}_v, \mathbf{e}_{(u,v)}, c_{\{u,v\}})$$

where $\Theta$ are learnable weights, $\mathbf{x}_u$, $\mathbf{x}_v$ and $\mathbf{e}_{(u,v)}$ are the node and edge features respectively, while $c_{\{u,v\}}$ is the capacity of the channel. While several choices are possible for $f_\Theta$, such as multi-layer perceptrons ( Rosenblatt (1957)) or Graph Neural Networks, we focus on Random Forests for this work given their simplicity and efficacy. In particular, our Random Forest (RF) model operates on the concatenation of the features of the source and destination nodes as well as the edge features:

$$\hat{p}_{(u,v)} = \mathrm{RF}(x_u \,||\, z_u \,||\, x_v \,||\, z_v \,||\, e_{(u,v)})$$

The model is trained using a Mean Squared Error loss.

### 2.7.2 NODE FEATURES

- **Node Feature Flags**
  0-1 vector indicating which of the features each node supports. For example, feature flag 19, the wumbo flag.

- **Capacity Centrality**
  The node's capacity divided by the network's capacity. This indicates how much of the network's capacity is incident to a node.

- **Fee Ratio**
  Ratio of the mean cost of a nodes outgoing fees to the mean cost of its incoming fees.

### 2.7.3 EDGE FEATURES

- **Time Lock Delta** The number of blocks a relayed payment is locked into an HTLC.

- **Min HTLC** The minimum amount this edge will route. (denominated in millisats)

- **Max HTLC msat** The maximum amount this edge will route. (denominated in millisats)

- **Fee Rate millimsat** Proportional fee to route a payment along an edge. (denominated in millimillisats)

- **Fee Base msat** Fixed fee to route a payment along an edge. (denominated in millisats)

### 2.7.4 POSITIONAL ENCODINGS

Positional encoding is essential for capturing the structural context of nodes, as graphs lack inherent sequential order. Utilizing eigenvectors of the graph Laplacian matrix as positional encodings provides a robust solution to this challenge. These eigenvectors highlight key structural patterns, enriching node features with information about the overall topology of the graphDwivedi et al. (2020). By integrating these spectral properties, machine learning models can effectively recognize and utilize global characteristics, enhancing performance in tasks like node classification and community detection.

### 2.7.5 CONCATENATED PREDICTION ML MODEL

Our model predicts $p_{(u,v)}$ as a function of the concatenation of the features of the source and destination nodes as well as edge features:

$$\hat{p}_{(u,v)} = \mathrm{RF}(x_u \,||\, x_v \,||\, e_{(u,v)})$$

### 2.7.6 MODEL TRAINING DETAILS AND PERFORMANCE

We set aside 10% of the observed $y_{(u,v)}$ as our test set and 10% as our validation set. The RF model trained for balance prediction has an MAE of 1.08, an R of 0.612 and $R^2$ of 0.365. For the BBS experiments, we use the predictions on the validation set (89 channels). For validation set channels, we feed the predictions of each tree in the ensemble (100) into a kernel density estimation (KDE) process to construct the search space PDF. Using these constructed PDFs, we compare BBS with basic binary search in measuring how many probes it would take to ascertain the balance of a given channel.

Table 2: Lightning Channel Probing Comparison

| $\epsilon$ | Percent Decrease | Basic Mean Steps | BBS Mean Steps |
|---|---|---|---|
| 128 | 3.07% | $14.26 \pm 2.26$ | $13.82 \pm 2.35$ |
| 256 | 3.31% | $13.26 \pm 2.26$ | $12.82 \pm 2.35$ |
| 512 | 3.57% | $12.26 \pm 2.26$ | $11.82 \pm 2.35$ |
| 1024 | 3.89% | $11.26 \pm 2.26$ | $10.82 \pm 2.35$ |
| 2048 | 4.27% | $10.26 \pm 2.26$ | $9.82 \pm 2.35$ |
| 4096 | 4.73% | $9.26 \pm 2.26$ | $8.82 \pm 2.35$ |
| 8192 | 5.03% | $8.26 \pm 2.26$ | $7.84 \pm 2.30$ |
| 16384 | 5.73% | $7.26 \pm 2.26$ | $6.84 \pm 2.29$ |

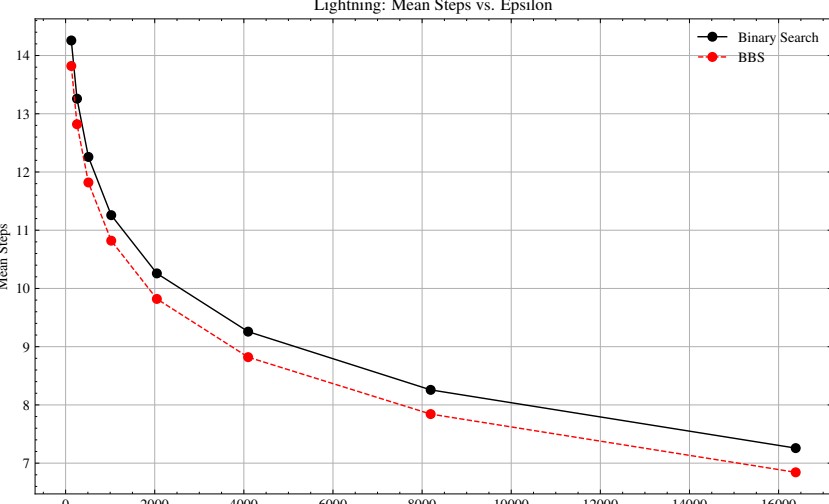

Figure 2: Lightning Probing Experiment: Basic vs. BBS Convergence Comparison

## 3 DISCUSSION

Our experimental results demonstrate the potential of Bayesian Binary Search (BBS) as a promising alternative to classical binary search/bisection, particularly in scenarios where the search space exhibits non-uniform distributions and is costly to access. The performance improvements observed in both simulated environments and real-world applications, such as probing payment channels in the Bitcoin Lightning Network, highlight the promise of BBS. However, it is important to consider these findings in a broader context and acknowledge the limitations and implications of our work.

### 3.1 THEORETICAL ALIGNMENT AND PERFORMANCE GAINS

The superior performance of BBS over classical binary search in non-uniform distributions aligns with our theoretical expectations. By leveraging probabilistic information about the search space, BBS can make more informed decisions at each step, leading to faster convergence on average. This is particularly evident in our simulated experiments with skewed distributions, where BBS consistently required fewer steps to locate the target. The ability of BBS to adapt to the underlying distribution of the search space represents an advance in the design of search algorithms, and a fusion of search theory and probabilistic machine learning/statistical learning.

### 3.2 REAL-WORLD APPLICATION AND IMPLICATIONS

In the context of the Bitcoin Lightning Network, the improvement in probing efficiency could have significant practical implications. Faster and more accurate channel capacity discovery can enhance routing algorithms, potentially leading to improved transaction success rates. Crucially, probes in the Lightning Network consume network-wide computational resources, as each probe requires

processing by multiple nodes along potential payment routes. By using BBS to enhance probe efficiency, we effectively reduce unnecessary network load, akin to removing spam from the system. This reduction in network overhead could lead to improved overall network performance and scalability.

### 3.3 LIMITATIONS AND CHALLENGES

Despite the promising results, several limitations of our study should be acknowledged:

1. **Computational Overhead:** The improved search efficiency of BBS comes at the cost of increased computational complexity, particularly in the PDF estimation step. This overhead may offset the reduction in search steps for certain applications, especially those dealing with small search spaces or requiring extremely fast operation.

2. **PDF Estimation Accuracy:** The performance of BBS is heavily dependent on the accuracy of the PDF estimation. In scenarios where the underlying distribution is highly complex or rapidly changing, the chosen PDF estimation method may struggle to provide accurate probabilities, potentially leading to suboptimal performance.

### 3.4 FUTURE RESEARCH DIRECTIONS

Our findings open up several exciting avenues for future research:

1. **Adaptive PDF Estimation:** Developing methods to dynamically adjust the PDF estimation technique based on the observed search space characteristics could further improve the robustness and efficiency of BBS.

2. **Theoretical Bounds:** While we provide empirical evidence of BBS's effectiveness, deriving tighter theoretical bounds on its performance under various distribution types would strengthen its theoretical foundation.

3. **Application to Other Domains:** Exploring the applicability of BBS to other areas such as database indexing, computational biology, or optimization algorithms could reveal new use cases and challenges.

### 3.5 CONCLUSION

Bayesian Binary Search represents a significant step towards more adaptive and efficient search algorithms. By incorporating probabilistic information, BBS demonstrates the potential to outperform classical binary search in non-uniform distributions, as evidenced by our experiments in both simulated and real-world scenarios. In the context of the Lightning Network, BBS not only improves search efficiency but also contributes to reducing unnecessary network load, potentially enhancing the overall performance and scalability of the system. The promising results open up new possibilities for optimizing search processes in various domains. As we continue to navigate increasingly complex and data-rich environments, algorithms like BBS that can adapt to the underlying structure of the problem space will become increasingly valuable. Future work in this direction has the potential to not only improve search efficiency but also to deepen our understanding of how probabilistic approaches can enhance fundamental algorithms in computer science and their real-world applications.

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

# A  APPENDIX

## A.1  BBS ON IMPERFECT SEARCH SPACE DENSITY ESTIMATION

The performance of BBS depends on the accuracy of the search space density estimation. To demonstrate the performance of BBS on varying degrees of accuracy in search space density estimation, we run experiments in which the estimated search space probability density function is a given Kullback-Leibler (KL) divergence from the target normal distribution of the search space.

We derive a new normal distribution with a specified KL divergence to a target distribution below:

Given:

- Target distribution: $N(\mu_1, \sigma_1^2)$
- Desired KL divergence: $D$

We want to find parameters $\mu_2$ and $\sigma_2$ for a new distribution $N(\mu_2, \sigma_2^2)$ such that:

$$KL(N(\mu_1, \sigma_1^2) \parallel N(\mu_2, \sigma_2^2)) = D$$

The KL divergence between two normal distributions is given by:

$$KL(N(\mu_1, \sigma_1^2) \parallel N(\mu_2, \sigma_2^2)) = \log \frac{\sigma_2}{\sigma_1} + \frac{\sigma_1^2 + (\mu_1 - \mu_2)^2}{2\sigma_2^2} - \frac{1}{2}$$

For simplicity, let's assume $\sigma_2 = \sigma_1$. Then the equation simplifies to:

$$D = \frac{(\mu_1 - \mu_2)^2}{2\sigma_1^2}$$

Solving for $\mu_2$:

$$D = \frac{(\mu_1 - \mu_2)^2}{2\sigma_1^2}$$

$$2D\sigma_1^2 = (\mu_1 - \mu_2)^2$$

$$\sqrt{2D\sigma_1^2} = \mid \mu_1 - \mu_2 \mid$$

$$\mu_2 = \mu_1 \pm \sqrt{2D\sigma_1^2}$$

We choose the positive solution:

$$\mu_2 = \mu_1 + \sqrt{2D\sigma_1^2}$$

Therefore, the parameters of the new distribution are:

$$\mu_2 = \mu_1 + \sqrt{2D\sigma_1^2}$$

$$\sigma_2 = \sigma_1$$

This ensures that:

$$KL(N(\mu_1, \sigma_1^2) \parallel N(\mu_2, \sigma_2^2)) = D$$

Table 3: $\mu = 0.0, \sigma = 1000.0, \epsilon=10$, N=200

| KLD | Percent Decrease | Basic Mean Steps | Bayesian Mean Steps |
|---|---|---|---|
| 0.0 | 9.40% | $10.00 \pm 0.00$ | $9.06 \pm 0.95$ |
| 0.05 | 8.85% | $10.00 \pm 0.00$ | $9.12 \pm 1.07$ |
| 0.1 | 7.70% | $10.00 \pm 0.00$ | $9.23 \pm 1.12$ |
| 0.15 | 6.95% | $10.00 \pm 0.00$ | $9.30 \pm 1.19$ |
| 0.2 | 6.35% | $10.00 \pm 0.00$ | $9.37 \pm 1.26$ |
| 0.25 | 6.15% | $10.00 \pm 0.00$ | $9.38 \pm 1.34$ |
| 0.3 | 4.90% | $10.00 \pm 0.00$ | $9.51 \pm 1.40$ |
| 0.35 | 4.35% | $10.00 \pm 0.00$ | $9.56 \pm 1.44$ |
| 0.4 | 3.00% | $10.00 \pm 0.00$ | $9.70 \pm 1.42$ |
| 0.45 | 2.25% | $10.00 \pm 0.00$ | $9.78 \pm 1.56$ |
| 0.5 | 1.20% | $10.00 \pm 0.00$ | $9.88 \pm 1.62$ |
| 0.55 | 0.60% | $10.00 \pm 0.00$ | $9.94 \pm 1.65$ |
| 0.6 | 0.60% | $10.00 \pm 0.00$ | $9.94 \pm 1.66$ |
| 0.65 | -0.20% | $10.00 \pm 0.00$ | $10.02 \pm 1.77$ |
| 0.7 | -0.90% | $10.00 \pm 0.00$ | $10.09 \pm 1.80$ |
| 0.75 | -1.90% | $10.00 \pm 0.00$ | $10.19 \pm 1.87$ |
| 0.8 | -2.75% | $10.00 \pm 0.00$ | $10.28 \pm 1.94$ |
| 0.85 | -3.35% | $10.00 \pm 0.00$ | $10.34 \pm 2.00$ |
| 0.9 | -4.10% | $10.00 \pm 0.00$ | $10.41 \pm 2.00$ |
| 0.95 | -4.85% | $10.00 \pm 0.00$ | $10.48 \pm 2.03$ |

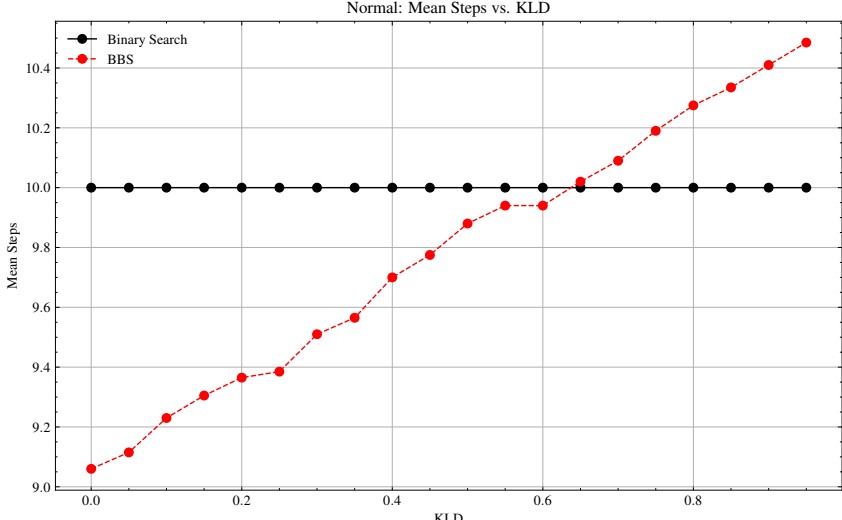

Figure 3: Basic vs BBS Convergence with KLD Separation of Estimated Search Space PDF and Actual PDF

Table 4: Bimodal Params, $\mu_1 = 0, \sigma_1 = 1000, \mu_2 = 4000, \sigma_2 = 1000, w_1 = 0.5$

| $\epsilon$ | Percent Decrease | Basic Mean Steps | Bayesian Mean Steps |
|---|---|---|---|
| 1 | 3.14% | $13.38 \pm 0.49$ | $12.96 \pm 0.99$ |
| 2 | 2.38% | $12.58 \pm 0.50$ | $12.28 \pm 0.90$ |
| 3 | 2.67% | $12.00 \pm 0.00$ | $11.68 \pm 0.79$ |
| 4 | 4.22% | $11.86 \pm 0.35$ | $11.36 \pm 0.72$ |
| 5 | 0.36% | $11.00 \pm 0.00$ | $10.96 \pm 0.78$ |
| 6 | 2.91% | $11.00 \pm 0.00$ | $10.68 \pm 0.82$ |
| 7 | 4.55% | $11.00 \pm 0.00$ | $10.50 \pm 0.74$ |
| 8 | 5.27% | $11.00 \pm 0.00$ | $10.42 \pm 0.64$ |
| 9 | 1.53% | $10.48 \pm 0.50$ | $10.32 \pm 0.51$ |
| 10 | 0.40% | $10.00 \pm 0.00$ | $9.96 \pm 0.78$ |
| 11 | 2.80% | $10.00 \pm 0.00$ | $9.72 \pm 0.81$ |
| 12 | 3.80% | $10.00 \pm 0.00$ | $9.62 \pm 0.75$ |
| 13 | 4.40% | $10.00 \pm 0.00$ | $9.56 \pm 0.76$ |
| 14 | 5.00% | $10.00 \pm 0.00$ | $9.50 \pm 0.71$ |
| 15 | 5.60% | $10.00 \pm 0.00$ | $9.44 \pm 0.67$ |
| 16 | 5.80% | $10.00 \pm 0.00$ | $9.42 \pm 0.64$ |
| 17 | 6.40% | $10.00 \pm 0.00$ | $9.36 \pm 0.56$ |
| 18 | 6.60% | $10.00 \pm 0.00$ | $9.34 \pm 0.52$ |
| 19 | -1.98% | $9.08 \pm 0.27$ | $9.26 \pm 0.56$ |
| 20 | 0.00% | $9.00 \pm 0.00$ | $9.00 \pm 0.78$ |
| 21 | 1.78% | $9.00 \pm 0.00$ | $8.84 \pm 0.84$ |
| 22 | 2.67% | $9.00 \pm 0.00$ | $8.76 \pm 0.74$ |
| 23 | 3.78% | $9.00 \pm 0.00$ | $8.66 \pm 0.75$ |
| 24 | 4.00% | $9.00 \pm 0.00$ | $8.64 \pm 0.75$ |
| 25 | 4.00% | $9.00 \pm 0.00$ | $8.64 \pm 0.75$ |
| 26 | 4.44% | $9.00 \pm 0.00$ | $8.60 \pm 0.76$ |
| 27 | 4.44% | $9.00 \pm 0.00$ | $8.60 \pm 0.76$ |
| 28 | 5.11% | $9.00 \pm 0.00$ | $8.54 \pm 0.76$ |
| 29 | 6.00% | $9.00 \pm 0.00$ | $8.46 \pm 0.68$ |
| 30 | 6.22% | $9.00 \pm 0.00$ | $8.44 \pm 0.67$ |
| 31 | 6.44% | $9.00 \pm 0.00$ | $8.42 \pm 0.64$ |
| 32 | 6.67% | $9.00 \pm 0.00$ | $8.40 \pm 0.61$ |

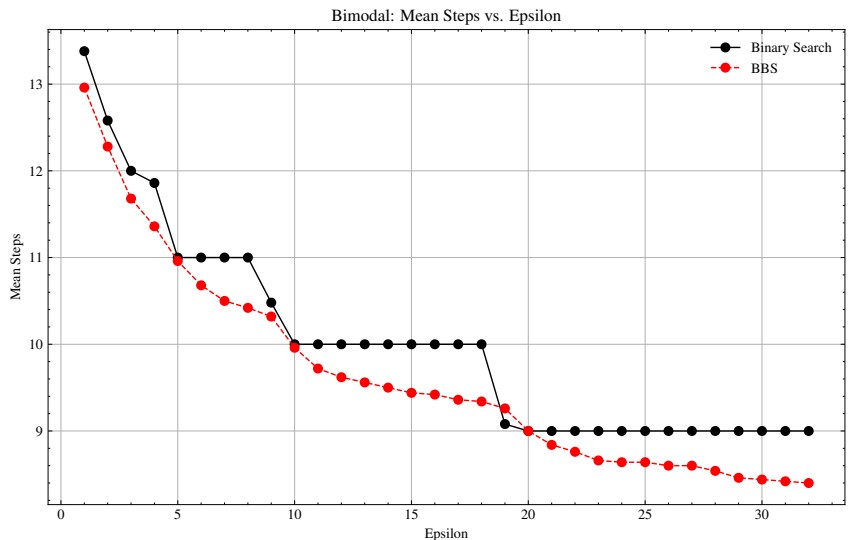

Figure 4: Bimodal Distribution: Basic vs. BBS Convergence Comparison

Table 5: Exponential Params, scale=10000

| $\epsilon$ | Percent Decrease | Basic Mean Steps | Bayesian Mean Steps |
|---|---|---|---|
| 1 | 12.86% | $16.87 \pm 0.34$ | $14.70 \pm 1.41$ |
| 2 | 13.03% | $16.00 \pm 0.00$ | $13.91 \pm 1.34$ |
| 3 | 14.40% | $15.55 \pm 0.50$ | $13.31 \pm 1.37$ |
| 4 | 13.07% | $15.00 \pm 0.00$ | $13.04 \pm 1.33$ |
| 5 | 15.10% | $15.00 \pm 0.00$ | $12.73 \pm 1.35$ |
| 6 | 17.17% | $15.00 \pm 0.00$ | $12.43 \pm 1.36$ |
| 7 | 13.01% | $14.03 \pm 0.16$ | $12.20 \pm 1.37$ |
| 8 | 13.75% | $14.00 \pm 0.00$ | $12.07 \pm 1.34$ |
| 9 | 14.39% | $14.00 \pm 0.00$ | $11.98 \pm 1.31$ |
| 10 | 15.54% | $14.00 \pm 0.00$ | $11.82 \pm 1.34$ |
| 11 | 16.89% | $14.00 \pm 0.00$ | $11.63 \pm 1.36$ |
| 12 | 18.11% | $14.00 \pm 0.00$ | $11.46 \pm 1.36$ |
| 13 | 19.00% | $14.00 \pm 0.00$ | $11.34 \pm 1.35$ |
| 14 | 13.83% | $13.05 \pm 0.23$ | $11.25 \pm 1.36$ |
| 15 | 14.27% | $13.00 \pm 0.00$ | $11.14 \pm 1.37$ |
| 16 | 14.62% | $13.00 \pm 0.00$ | $11.10 \pm 1.37$ |
| 17 | 15.00% | $13.00 \pm 0.00$ | $11.05 \pm 1.34$ |
| 18 | 15.38% | $13.00 \pm 0.00$ | $11.00 \pm 1.32$ |
| 19 | 15.65% | $13.00 \pm 0.00$ | $10.96 \pm 1.32$ |
| 20 | 16.42% | $13.00 \pm 0.00$ | $10.87 \pm 1.34$ |
| 21 | 17.35% | $13.00 \pm 0.00$ | $10.74 \pm 1.36$ |
| 22 | 18.19% | $13.00 \pm 0.00$ | $10.63 \pm 1.36$ |
| 23 | 18.50% | $13.00 \pm 0.00$ | $10.60 \pm 1.37$ |
| 24 | 19.08% | $13.00 \pm 0.00$ | $10.52 \pm 1.34$ |
| 25 | 19.81% | $13.00 \pm 0.00$ | $10.43 \pm 1.35$ |
| 26 | 20.15% | $13.00 \pm 0.00$ | $10.38 \pm 1.34$ |
| 27 | 20.77% | $13.00 \pm 0.00$ | $10.30 \pm 1.34$ |
| 28 | 15.42% | $12.13 \pm 0.34$ | $10.26 \pm 1.36$ |
| 29 | 14.92% | $12.00 \pm 0.00$ | $10.21 \pm 1.37$ |
| 30 | 15.42% | $12.00 \pm 0.00$ | $10.15 \pm 1.37$ |
| 31 | 15.58% | $12.00 \pm 0.00$ | $10.13 \pm 1.37$ |
| 32 | 15.75% | $12.00 \pm 0.00$ | $10.11 \pm 1.37$ |

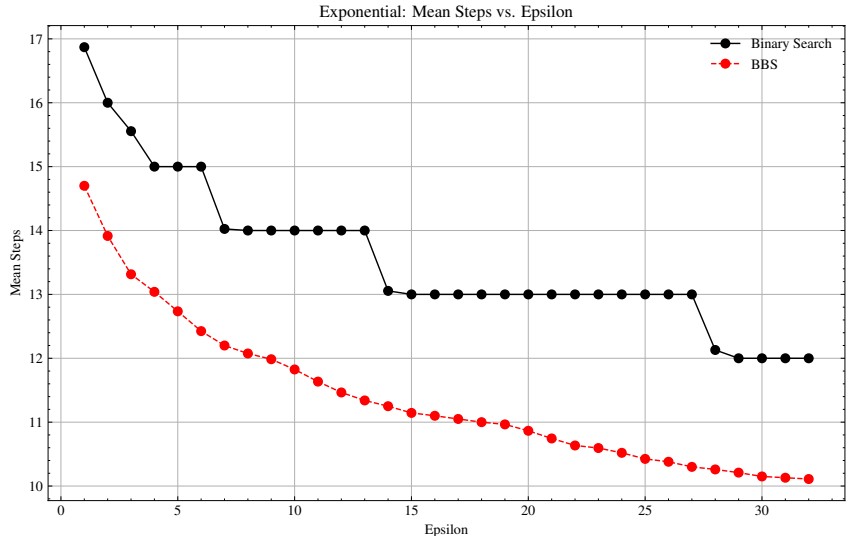

Figure 5: Exponential Distribution: Basic vs. BBS Convergence Comparison

Table 6: Beta Params a=1, b=5, scale=10000 n=500

| $\epsilon$ | Percent Decrease | Basic Mean Steps | Bayesian Mean Steps |
|---|---|---|---|
| 1 | 4.26% | $13.25 \pm 0.43$ | $12.68 \pm 0.92$ |
| 2 | 4.36% | $12.39 \pm 0.49$ | $11.85 \pm 0.89$ |
| 3 | 5.43% | $12.00 \pm 0.00$ | $11.35 \pm 0.79$ |
| 4 | 6.20% | $11.62 \pm 0.49$ | $10.90 \pm 0.85$ |
| 5 | 4.04% | $11.00 \pm 0.00$ | $10.56 \pm 0.82$ |
| 6 | 5.44% | $11.00 \pm 0.00$ | $10.40 \pm 0.76$ |
| 7 | 6.67% | $11.00 \pm 0.00$ | $10.27 \pm 0.73$ |
| 8 | 9.42% | $11.00 \pm 0.00$ | $9.96 \pm 0.84$ |
| 9 | 3.65% | $10.08 \pm 0.27$ | $9.71 \pm 0.85$ |
| 10 | 4.24% | $10.00 \pm 0.00$ | $9.58 \pm 0.85$ |
| 11 | 5.14% | $10.00 \pm 0.00$ | $9.49 \pm 0.81$ |
| 12 | 5.84% | $10.00 \pm 0.00$ | $9.42 \pm 0.77$ |
| 13 | 6.28% | $10.00 \pm 0.00$ | $9.37 \pm 0.74$ |
| 14 | 6.74% | $10.00 \pm 0.00$ | $9.33 \pm 0.71$ |
| 15 | 7.44% | $10.00 \pm 0.00$ | $9.26 \pm 0.72$ |
| 16 | 9.50% | $10.00 \pm 0.00$ | $9.05 \pm 0.81$ |
| 17 | 11.44% | $10.00 \pm 0.00$ | $8.86 \pm 0.86$ |
| 18 | 4.50% | $9.15 \pm 0.36$ | $8.74 \pm 0.86$ |
| 19 | 3.80% | $9.00 \pm 0.00$ | $8.66 \pm 0.86$ |
| 20 | 4.56% | $9.00 \pm 0.00$ | $8.59 \pm 0.86$ |
| 21 | 5.09% | $9.00 \pm 0.00$ | $8.54 \pm 0.83$ |
| 22 | 5.64% | $9.00 \pm 0.00$ | $8.49 \pm 0.82$ |
| 23 | 5.98% | $9.00 \pm 0.00$ | $8.46 \pm 0.79$ |
| 24 | 6.31% | $9.00 \pm 0.00$ | $8.43 \pm 0.77$ |
| 25 | 6.53% | $9.00 \pm 0.00$ | $8.41 \pm 0.76$ |
| 26 | 6.84% | $9.00 \pm 0.00$ | $8.38 \pm 0.75$ |
| 27 | 7.24% | $9.00 \pm 0.00$ | $8.35 \pm 0.72$ |
| 28 | 7.47% | $9.00 \pm 0.00$ | $8.33 \pm 0.70$ |
| 29 | 7.58% | $9.00 \pm 0.00$ | $8.32 \pm 0.69$ |
| 30 | 7.84% | $9.00 \pm 0.00$ | $8.29 \pm 0.68$ |
| 31 | 8.36% | $9.00 \pm 0.00$ | $8.25 \pm 0.71$ |
| 32 | 10.36% | $9.00 \pm 0.00$ | $8.07 \pm 0.80$ |

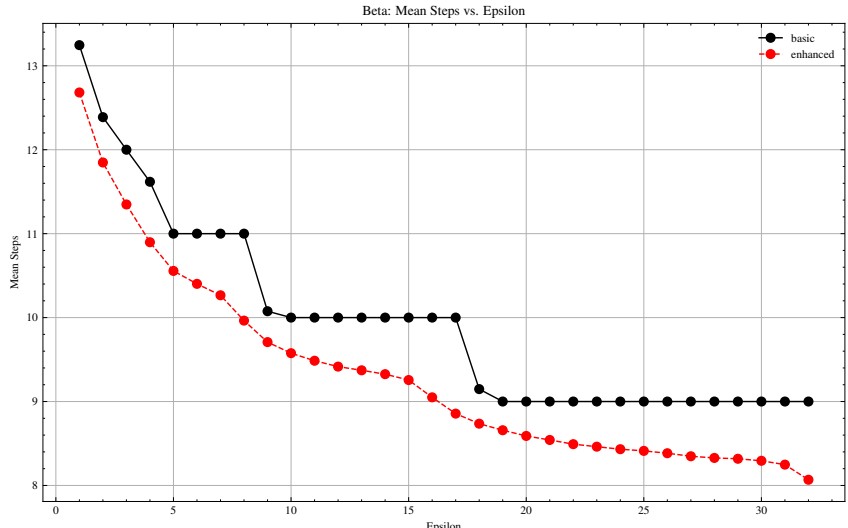

Figure 6: Beta Distribution: Basic vs. BBS Convergence Comparison

Table 7: Lognorm Params s=1, scale=10000, n=500

| $\epsilon$ | Percent Decrease | Basic Mean Steps | Bayesian Mean Steps |
|---|---|---|---|
| 1 | 21.08% | $19.53 \pm 0.50$ | $15.41 \pm 1.90$ |
| 2 | 22.52% | $18.78 \pm 0.41$ | $14.55 \pm 1.89$ |
| 3 | 21.72% | $18.00 \pm 0.00$ | $14.09 \pm 1.86$ |
| 4 | 24.10% | $18.00 \pm 0.00$ | $13.66 \pm 1.89$ |
| 5 | 23.51% | $17.46 \pm 0.50$ | $13.36 \pm 1.89$ |
| 6 | 22.42% | $17.00 \pm 0.00$ | $13.19 \pm 1.85$ |
| 7 | 23.78% | $17.00 \pm 0.00$ | $12.96 \pm 1.82$ |
| 8 | 25.28% | $17.00 \pm 0.00$ | $12.70 \pm 1.90$ |
| 9 | 26.40% | $17.00 \pm 0.00$ | $12.51 \pm 1.91$ |
| 10 | 26.44% | $16.85 \pm 0.36$ | $12.39 \pm 1.91$ |
| 11 | 23.15% | $16.00 \pm 0.00$ | $12.30 \pm 1.88$ |
| 12 | 23.64% | $16.00 \pm 0.00$ | $12.22 \pm 1.86$ |
| 13 | 24.19% | $16.00 \pm 0.00$ | $12.13 \pm 1.85$ |
| 14 | 24.68% | $16.00 \pm 0.00$ | $12.05 \pm 1.82$ |
| 15 | 25.88% | $16.00 \pm 0.00$ | $11.86 \pm 1.87$ |
| 16 | 26.69% | $16.00 \pm 0.00$ | $11.73 \pm 1.89$ |
| 17 | 27.36% | $16.00 \pm 0.00$ | $11.62 \pm 1.93$ |
| 18 | 27.84% | $16.00 \pm 0.00$ | $11.55 \pm 1.93$ |
| 19 | 28.34% | $16.00 \pm 0.00$ | $11.47 \pm 1.90$ |
| 20 | 28.69% | $16.00 \pm 0.00$ | $11.41 \pm 1.90$ |
| 21 | 27.63% | $15.71 \pm 0.45$ | $11.37 \pm 1.90$ |
| 22 | 24.44% | $15.00 \pm 0.00$ | $11.33 \pm 1.89$ |
| 23 | 24.84% | $15.00 \pm 0.00$ | $11.27 \pm 1.88$ |
| 24 | 25.07% | $15.00 \pm 0.00$ | $11.24 \pm 1.87$ |
| 25 | 25.37% | $15.00 \pm 0.00$ | $11.19 \pm 1.86$ |
| 26 | 25.65% | $15.00 \pm 0.00$ | $11.15 \pm 1.85$ |
| 27 | 25.93% | $15.00 \pm 0.00$ | $11.11 \pm 1.84$ |
| 28 | 26.09% | $15.00 \pm 0.00$ | $11.09 \pm 1.84$ |
| 29 | 26.44% | $15.00 \pm 0.00$ | $11.03 \pm 1.81$ |
| 30 | 27.21% | $15.00 \pm 0.00$ | $10.92 \pm 1.85$ |
| 31 | 27.95% | $15.00 \pm 0.00$ | $10.81 \pm 1.88$ |
| 32 | 28.24% | $15.00 \pm 0.00$ | $10.76 \pm 1.89$ |

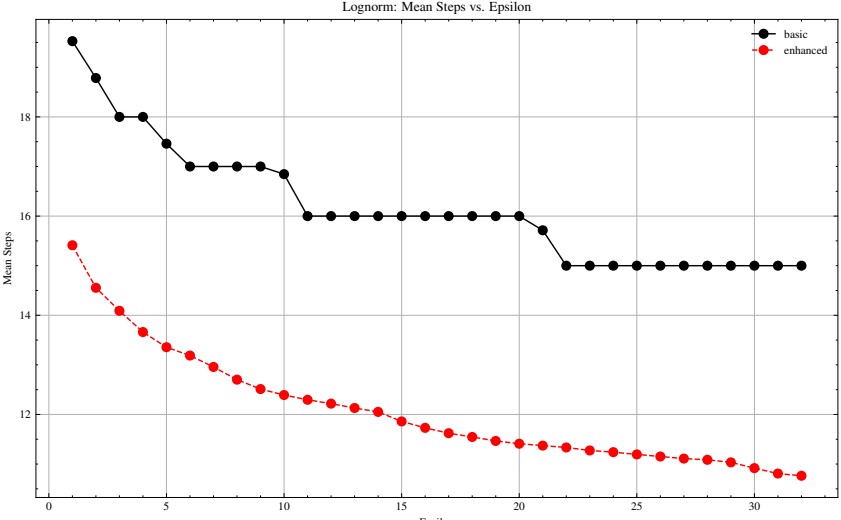

Figure 7: Lognorm Distribution: Basic vs. BBS Convergence Comparison

## A.2 BINARY SEARCH TREE VISUALIZATION COMPARISON

To visualize the difference between the execution of BBS and standard binary search in a normally distributed search space $\mu = 5$, $\sigma = 1.15$, we show binary search trees (BSTs) for basic binary search and BBS over 10000 iterations of the experiment. We observe a significantly higher percentage of BBS searches terminating at a depth of 2, where all basic searches in this experiment terminate at depth 3 or 4. The bracket represents the search interval, and the number below for internal nodes represents the median of that search interval. The number below the bracket for the green leaf nodes represent how many times the search terminated at that leaf.

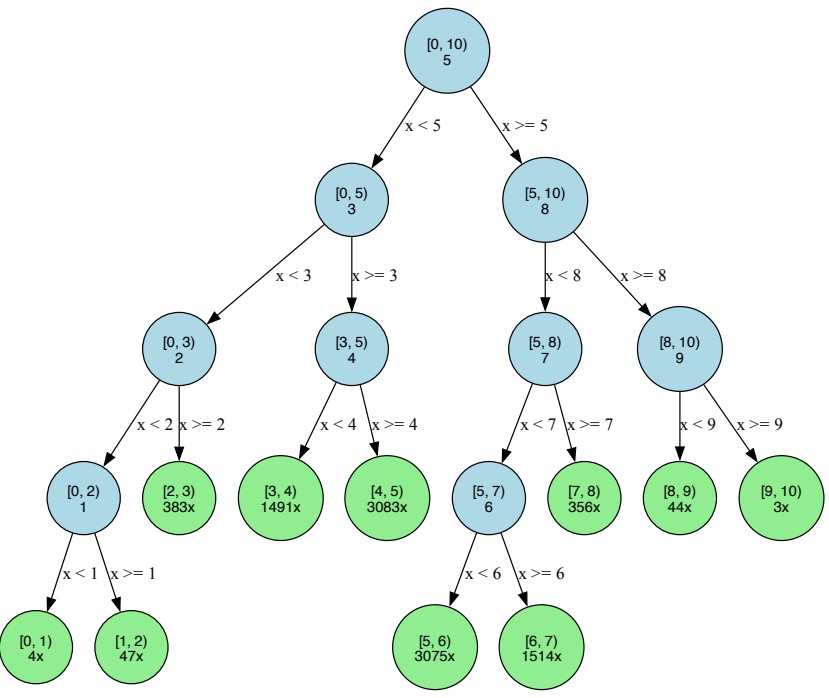

Figure 8: Binary Search Tree

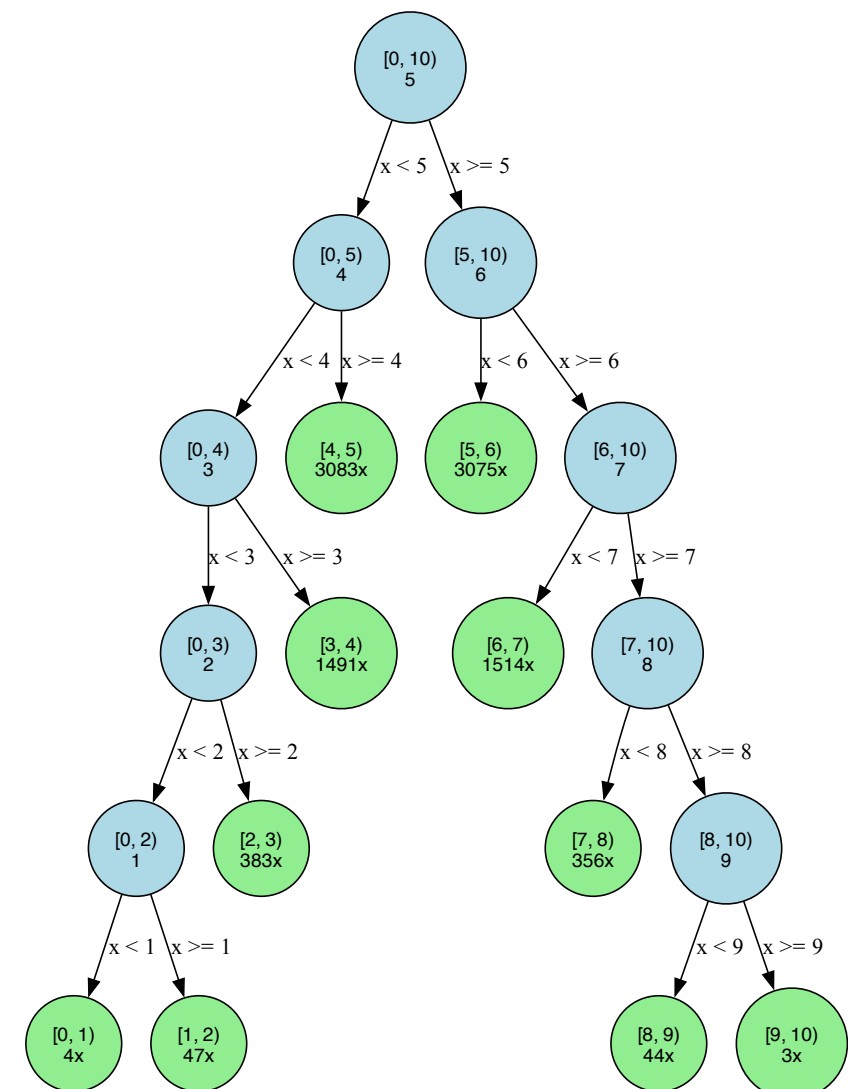

Figure 9: Bayesian Binary Search Tree

## A.3 THEORETICAL ANALYSIS

**Terms:**
Domain/Search Space: $x \in [a_0, b_0]$, where $a_0, b_0 \in \mathbb{R}$ and $a_0 \leq b_0$
Unknown Target: $t \in [a_0, b_0]$

Sign Function: $s(x) = \begin{cases} 1 & x > t \\ -1 & x \leq t \end{cases}$

Boundary Conditions: $s(a_0) = -1$, $s(b_0) = 1$
Tolerance: $\epsilon > 0$

### PROOF OF CONVERGENCE OF BISECTION METHOD ON DOMAIN SPACE

### ALGORITHM DESCRIPTION

The bisection method starts with an initial interval $[a_0, b_0]$ such that $s(a_0) = -1$ and $s(b_0) = 1$. The interval is halved at each step, and the new interval is chosen based on the sign of the function $s$ at the midpoint. This process is repeated until the length of the interval is less than or equal to a specified tolerance $\epsilon$, where $\epsilon > 0$.

Let the length of the initial interval be $N = b_0 - a_0$. At each iteration, the interval is halved, so after $k$ iterations, the length of the interval becomes:

$$\text{Length of interval after } k \text{ iterations} = \frac{N}{2^k}$$

STOPPING CONDITION

The method stops when the length of the interval is less than or equal to $\epsilon$, i.e., when:

$$\frac{N}{2^k} \leq \epsilon$$

Solving for $k$, we get:

$$2^k \geq \frac{N}{\epsilon}$$

Taking the logarithm (base 2) of both sides:

$$k \geq \log_2\left(\frac{N}{\epsilon}\right)$$

Thus, the number of iterations $k$ required to achieve the desired tolerance $\epsilon$ is:

$$k = \left\lceil \log_2\left(\frac{N}{\epsilon}\right) \right\rceil$$

PROOF OF CONVERGENCE OF BISECTION METHOD ON PROBABILITY DENSITY SPACE

ALGORITHM DESCRIPTION

Given the same framework of terms as above, plus:

$$\underset{\text{percentile}}{p} = F(x) = CDF(x) = \int_{-\infty}^{x} pdf(x)dx, p \in (0,1) \tag{2}$$

$$\underset{\text{domain value}}{x} = Q(p) = F^{-1}(p) \tag{3}$$

Bisecting on probability density space is similar to the above method where we bisect on the domain space.

The key difference is that rather than bisecting the domain space with an initial interval of $[a_0, b_0]$, we will bisect probability space with an initial interval of $(0,1)$, i.e. $(p_{lo_0}, p_{hi_0})$ where $s(Q(p_{lo_0})) = -1$ and $s(Q(p_{hi_0})) = 1$. Our search space then becomes $p \in (0,1)$, and at each iteration we will cut the percentile interval in half.

Our stopping condition then becomes: $Q(p_{hi_n}) - Q(p_{lo_n}) \leq \epsilon$

STOPPING CONDITION

Define the length of the interval as $p_{hi_n} - p_{lo_n}$. At iteration 0, $1 - 0 = 2^0$. At iteration 1, the interval could be $(0, 0.5]$ or $[0.5, 1)$, where the length of both ranges is $2^1$. Because the percentile interval gets cut in half at each iteration, we know that the length of the interval at iteration $k$ is $2^{-k}$. So at iteration $k$, the interval is equivalent to $[p_{lo_k}, p_{lo_k} + 2^{-k}]$. Note that the left or right side of the interval may remain open if $p_{lo}$ remains 0 or $p_{hi}$ remains 1 throughout the duration of the algorithm, respectively.

We can then rewrite our stopping condition of $Q(p_{hi_n}) - Q(p_{lo_n}) \leq \epsilon$ as $Q(p_{lo_n} + 2^{-n}) - Q(p_{lo_n}) \leq \epsilon$.

To prove that this algorithm converges, we know that as $n$ approaches $\infty$, $2^{-n}$ approaches 0. Thus, as $n$ approaches $\infty$, the length of the interval approaches 0. Therefore, because $\epsilon > 0$, we know this algorithm converges.

The rate of convergence depends on the probability distribution function.

The algorithm is described below:

---

**Algorithm 6:** Number of Steps Required for Bayesian Binary Search Convergence

1 Num-Steps $(Q, s, \epsilon)$
2 $i \leftarrow 0$
3 $(p_{\text{lo}}, p_{\text{hi}}) \leftarrow (0, 1)$
4 **while** $Q(p_{hi}) - Q(p_{lo}) > \epsilon$ **do**
5     $p_{\text{mid}} \leftarrow \frac{p_{\text{hi}} + p_{\text{lo}}}{2}$
6     $m \leftarrow Q(p_{\text{mid}})$
7     **if** $s(m) = -1$ **then**
8         $p_{\text{lo}} \leftarrow p_{\text{mid}}$
9     **else**
10         $p_{\text{hi}} \leftarrow p_{\text{mid}}$
11     $i \leftarrow i + 1$
12 **return** $i$

---

PROVING WORST CASE TIME COMPLEXITY OF BBS ON NORMAL DISTRIBUTION

The worst case of BBS on the normal distribution is when the target $t$ approaches the left or right tail of the normal distribution.

The proofs below assume that the target approaches $Q(0)$. However, the same methods can be trivially applied to $t \to Q(1)$.

Given that the length of the probability interval is $2^{-k}$ at iteration $k$, if target $t \to Q(0)$ then $\mathcal{N}_{\text{conv}}$ can be found via:

$$\mathcal{N}_{\text{conv}} = \underset{n}{\text{argmin}} \sum_{i=1}^{n} \left( Q(2^{-(i-1)}) - Q(2^{-i}) \right) \leq Q(1) - \epsilon$$

$$= \underset{n}{\text{argmin}} \, Q(2^{-n}) \leq \epsilon$$

After the first iteration, $p_{hi_1}$ becomes $2^{-1}$ and is repeatedly halved on each subsequent iteration.

At each iteration $k$, the searchable domain space is reduced by the quantity of the domain space to the right of $Q(p_{hi_k})$.

Subtracting the length of the domain space by the summation of the domain space reduced up to iteration $k$ is equal to the length of the interval in terms of the domain space. Technically, $Q(1)$ extends to positive infinity for the normal distribution, however it is canceled out in the proof below. The proof below proves that the summation is equivalent to the simplified expression below it.

**Proof by Induction**

Let $P(n)$ be the statement
$Q(1) - \sum_{i=1}^{n} \left( Q(2^{-(i-1)}) - Q(2^{-i}) \right) = Q(2^{-n})$ for all positive integers $n$.

**Basis step:** $P(1)$ is true.

$$P(1) = Q(1) - \sum_{i=1}^{1} \left( Q(2^{-(i-1)}) - Q(2^{-i}) \right)$$

$$= Q(1) - \left( Q(1) - Q(2^{-1}) \right)$$

$$= Q(2^{-1})$$

**Inductive step:** Assume $P(k)$ is true for some integer $k \geq 1$. Now we show $P(k+1)$ must also be true:

$$P(1) = Q(1) - \sum_{i=1}^{k+1} \left( Q(2^{-(i-1)}) - Q(2^{-i}) \right)$$

$$= Q(1) - \left[ \sum_{i=1}^{k} \left( Q(2^{-(i-1)}) - Q(2^{-i}) \right) + \left( Q(2^{-k}) - Q(2^{-(k+1)}) \right) \right]$$

$$= Q(2^{-k}) - Q(2^{-k}) + Q(2^{-(k+1)})$$

$$= Q(2^{-(k+1)})$$

$\therefore$ We have shown using the principle of mathematical induction that $P(n)$ is true for all positive integers $n$.

To prove $t \to Q(0)$ and $t \to Q(1)$ are the worst case in terms of time complexity for targets in the normal distribution, we will prove that no other target takes longer to converge.

**Theorem A.1.** *For a standard normal distribution with CDF $F(x)$ and quantile function $Q(x) = F^{-1}(x)$, the number of binary search steps required to achieve convergence within precision $\epsilon$ is maximized when the target approaches $Q(0)$ or $Q(1)$.*

*Proof.* We proceed by establishing several key properties of the quantile function and showing how they determine the convergence behavior of binary search.

Let us first establish our framework. For any target $t = Q(p)$ where $p \in (0,1)$, the binary search algorithm maintains an interval $[p_{\text{lo}}, p_{\text{hi}}]$ in probability space, corresponding to the interval $[Q(p_{\text{lo}}), Q(p_{\text{hi}})]$ in the domain space.

**Lemma A.2.** *The derivative of the quantile function $Q(x)$ is strictly increasing in magnitude as $\mid x \mid$ increases toward the tails of the distribution.*

*Proof of Lemma.* The derivative of the quantile function can be expressed as:

$$Q'(x) = \frac{1}{F'(Q(x))} = \frac{1}{\text{pdf}(Q(x))} \tag{4}$$

For the standard normal distribution:

$$\text{pdf}(x) = \frac{1}{\sqrt{2\pi}} e^{-x^2/2} \tag{5}$$

Therefore:

$$Q'(x) = \sqrt{2\pi} e^{Q(x)^2/2} \tag{6}$$

Since $Q(x)$ is monotonic and $\mid Q(x) \mid \to \infty$ as $x \to 0$ or 1, $Q'(x)$ strictly increases as $x$ approaches 0 or 1 starting from $x = 0.5$. $\square$

Now, consider the binary search algorithm at iteration $i$. The width of the interval in probability space is $2^{-i}$. Let $[a, a + 2^{-i}]$ be any such interval. The corresponding width in domain space is:

$$\mid Q(a + 2^{-i}) - Q(a) \mid = \int_a^{a+2^{-i}} Q'(x) dx \tag{7}$$

By the mean value theorem:

$$\mid Q(a + 2^{-i}) - Q(a) \mid = 2^{-i} Q'(c) \tag{8}$$

for some $c \in (a, a + 2^{-i})$.

Due to the monotonicity of $Q'(x)$ established in the lemma, this width is maximized when the interval $[a, a + 2^{-i}]$ is positioned as close as possible to either 0 or 1.

When the target approaches $Q(0)$ or $Q(1)$:

    1. The interval $[p_{\text{lo}}, p_{\text{hi}}]$ always includes one of the endpoints (0 or 1)

    2. The domain space interval $[Q(p_{\text{lo}}), Q(p_{\text{hi}})]$ is therefore maximized at each iteration

For convergence, we require:

$$\mid Q(p_{\text{hi}}) - Q(p_{\text{lo}}) \mid \leq \epsilon \tag{9}$$

The number of steps required to achieve this condition is determined by the width of the interval in domain space. Since this width is maximized when targeting $Q(0)$ or $Q(1)$, these targets require the maximum number of steps for convergence.

More formally, for any target $t = Q(p)$ where $p \in (0,1)$, let $N_{\text{conv}}(p)$ be the number of steps required for convergence. Then:

$$N_{\text{conv}}(p) \leq \max\{N_{\text{conv}}(0), N_{\text{conv}}(1)\} \tag{10}$$

Therefore, $Q(0)$ and $Q(1)$ represent the worst-case scenarios for convergence of the binary search algorithm. $\square$

### A.3.1 ADDITIONAL EXPERIMENTS WITH GPR DENSITY ESTIMATION AND RUNTIME APPROXIMATION

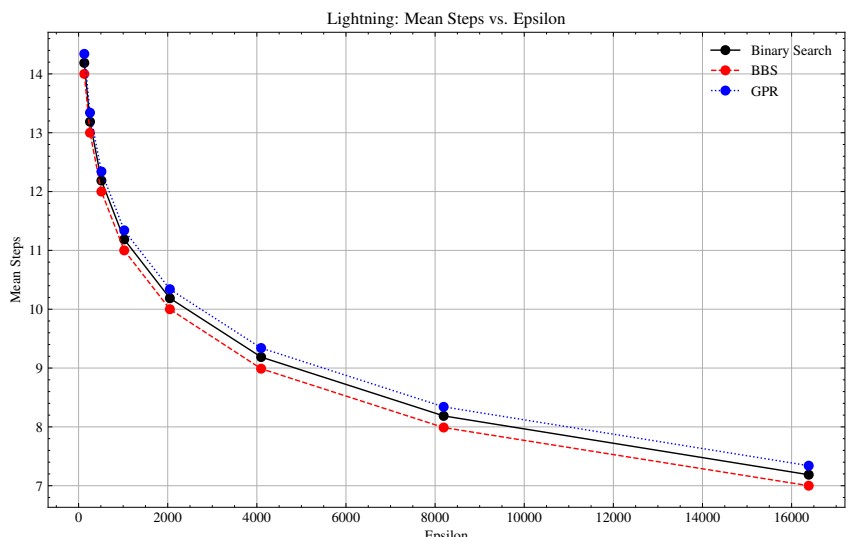

Figure 10: Lightning Probing Experiment: Basic vs. BBS Convergence Comparison

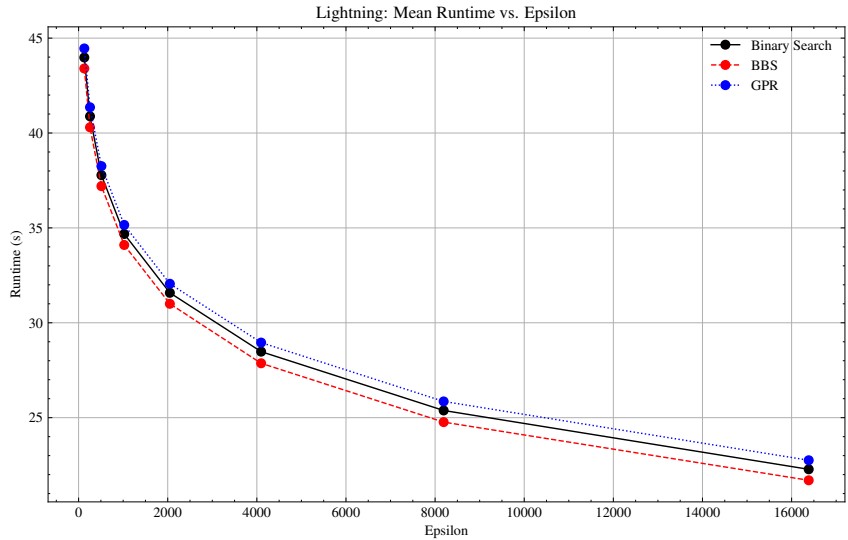

Figure 11: Lightning Probing Experiment: Basic vs. BBS Convergence Comparison (Runtime)

