# OpenReview forum: "Bayesian Binary Search"
_ICLR.cc/2025/Conference — Submitted to ICLR 2025_

### Official Review · Reviewer_Fj8u · 2024-10-20

**Soundness:** 2
**Presentation:** 2
**Contribution:** 2
**Rating:** 5
**Confidence:** 4

**Summary:**

The authors revisit bisection by introducing a probability density over the search interval rather than the usual uniform assumption. The main change is to cut where the probability reaches 50% rather than in the middle of the search interval. To estimate the required probability density, several alternatives are proposed. Then some simulation experiments are proposed.

**Strengths:**

- The experiments show some improvements over a simple baseline.
- A realistic application is considered: probing channel balances in the Bitcoin Lightning Network.

**Weaknesses:**

- The problem definition lacks the assumptions: for instance, bisection can only work if the underlying function is monotonous.  Otherwise, the problem may be of contour or level set estimation.
- It is not possible to separate the contribution from what already exists in the literature.
- The comparison is against a single simple baseline. Why not including comparisons with some of the cited references, like Waeber et al., or Jedynak et al?

**Questions:**

Could you clarify the assumptions and highlight better the contribution?
Under which conditions is the search provably successful? How does it depend on the estimated pdf error?

How is the variance over repeated runs zero in some tables? Perhaps the problem is too simple?

For Bayesian optimization: see e.g., Garnett, R. (2023). Bayesian optimization. Cambridge University Press. And for other probabilistic bisection algorithms:
- Rodriguez, S., & Ludkovski, M. (2020). Probabilistic bisection with spatial metamodels. European Journal of Operational Research, 286(2), 588-603.
- Frazier, P. I., Henderson, S. G., & Waeber, R. (2019). Probabilistic bisection converges almost as quickly as stochastic approximation. Mathematics of Operations Research, 44(2), 651-667.

Typos:
- specifoed
- Maxmimum

---

> ### Author Response · Authors · 2024-11-29
>
> We want to thank the reviewer for the thoughtful review and for providing the opportunity to improve the submission with this feedback. We have worked hard in the past couple of weeks to address this feedback, and update our manuscript as outlined below!
>
> Regarding the assumption addressed as a weakness in the paper, we clarify in the revised manuscript that the sign function here (Lightning channel balance oracle) is monotonic, and thus formally allows us to treat this as bisection.
>
> The probabilistic bisection literature mentioned (Waeber, Jedynak) is for noisy response problems (noisy oracle, so this is not the same problem we are tackling where we assume we are able to get noise free observations from the oracle, and thus we believe benchmarking to these algorithms is inappropriate.
>
> Regarding the weaknesses surrounding the lack of optimality with respect to the median split strategy, and the convergence bounds for imperfect PDF estimation, we agree that the lack of theoretic analysis is a bit strange for a paper on search algorithms. The initial reasoning for not including this analysis was that we present a general framework in which search space density estimation can be performed, and thus we did not want to make assumptions on distributions and accuracy when conducting this analysis as such analysis would heavily rely on those assumptions. To address this weakness however, we have now added a theoretical analysis section to the appendix in which we provide a convergence proof for the generalized bisection of density space with a tolerance. Additionally, we have added a convergence proof for the worst case convergence of bisecting density space on a perfectly approximated normal distribution.
>
> We expect the theoretical analysis of BBS to be quite an interesting topic for research in the coming years and are looking forward to learning more/collaborating with people that may be interested in that in the future.
>
> In our use case in the real world, we are not all constrained by the time to compute this search space density, but are constrained during the actual search process itself, by time, liquidity, and network throughput (max htlc limit). In fact we could precompute the binary search tree beforehand, and just traverse it during the search process.  We have updated the revised manuscript to include real probe times (3.1 seconds on average), along with density estimation time (0.18 seconds on average) to provide more context.
>
> The variance is 0 in some tables because of the experimental formulation of bisection with an unknown target. The oracle does not tell us when we have found the target exactly, but rather when we are within the range of the target via the remaining bracket size in the domain space. In standard bisection, the domain space is halved at each iteration, so the number of steps to reach a bracket size <= the tolerance is not dependent on where the target is and should be 0. The reason the variance is not 0 for some tolerances is because of our use of rounding to find the next median (e.g. a median of 11.5 would have our next probe be 12). In contrast, the target location does matter for BBS because the amount of the domain space removed from the search space at each iteration is dependent on the density of probability for the domain space surrounding it (e.g. in a normal distribution, if the bounds are p=0.5 and p=0.75, with a median p=0.625, more of the domain space would be cut if the search target is above $F^{-1}(0.625)$ because there is less density of values to the right).

---

> > ### Comment · Reviewer_Fj8u · 2024-12-02
> >
> > Thank you for replying to my comments, I appreciate the addition of a theoretical section and a clarification of the assumptions. Nevertheless, the empirical section would still benefit from additional comparisons to the existing literature. Even if some possible baselines handle noisy observations, they would work with a very small noise variance. Doing the reverse is usually more difficult.

---

> ### Author Response · Authors · 2024-12-02
>
> We thank the reviewer for the quick response!
>
> The PBA outlined and designed specifically for noisy observations reduces to the standard bisection algorithm when the noise is assumed to be 0 and a default uniform prior is selected, thus we still believe this to be a strong benchmark. For the simulation experiments, PBA without any noise reduces to bisecting in density space of the assumed prior (uniform in the literature most commonly, and thus equivalent to bisection), and here we believe we contribute empirical results that provide motivation for the kinds of gains one can achieve with supervised/unsupervised density estimation step since the PBA papers were not written to address noise free oracle responses.
>
> [From Waeber's paper "In all empirical examples we start the PBA with a uniform prior distribution f0."](https://arxiv.org/pdf/1807.00095)
>
> Additionally, working backwards from our Lightning Network real use case, we needed to critically add a density estimation step that is not inherent to PBA, since we would like to take advantage of our supervised machine learning model (and the data that allows us train that model) to enhance the search efficiency. We believe this perspective of providing a framework that includes flexible density estimation, including machine learning, is core to the contribution in this work.

---

### Official Review · Reviewer_XdTs · 2024-10-27

**Soundness:** 3
**Presentation:** 3
**Contribution:** 3
**Rating:** 6
**Confidence:** 4

**Summary:**

This paper introduces Bayesian Binary Search (BBS), a probabilistic approach compared to the traditional binary search, and it aims to improve search efficiency when the target distribution is non-uniform and the probing is expensive. BBS modifies the traditional binary search by replacing the midpoint splitting with median splitting w.r.t an estimated PDF.

Contributions include
1. Novel algorithm framework: the formulation of BBS integrates PDFs into binary search, and various of PDF estimation methods are supported in this framework. This framework promises better efficiency for non-uniform distribution, while reduces to traditional binary search when the distribution is uniform.
2. BBS demonstrates robust performance through KL divergence simulations, which shows graceful degradation as PDF estimation accuracy decreases
3. Showcase a real-world application in Bitcoin Lightning Network. The development of random forest PDF estimation methods using network features is itself valuable for practical implementations with BBS.

**Strengths:**

Originality
1. Creatively combines classical binary search with probabilistic methods and introduces a flexible framework for incorporating various of PDF estimation methods
2. Provides a new perspective on search optimization through statistical learning

Quality
1. Clear mathematical formulation and intuitive algorithm design
2. Rigorous analysis of robustness of BBS through KL divergence degradation simulation

Clarity
1. The paper is well structured and the clarity is pretty good

Significance
1. Interesting application to Bitcoin Lightning Network
2. This proposed framework can be easily implemented using various existing ML tools, and its significant performance improvements make it highly practical

**Weaknesses:**

1. Lack the theoretic analysis of optimality of median split strategy
2. Lack theoretic analysis for convergence bounds with imperfect PDF estimation
3. The paper mentioned a few methods for PDF estimation (RF, GPR, BNN, etc.), however, only RF was used for the Bitcoin Lighting Network problem. Since PDF estimation is a crucial piece in the algorithm framework, it is necessary to offer comparative analysis among different PDF estimation approaches, as well as discussion of the trade-offs.
4. Simulation study covered only the three basic distributions, but for real-world applications, it is common that the distribution is heavy-tailed or complex. I would expect more comprehensive analysis over real-world distribution patterns

**Questions:**

1. Beyond Lightning Network, can you elaborate on other applications you think would benefit most from BBS?
2. What characteristics make an application suitable for BBS?
3. Other than median splitting strategy, have you considered other strategies in probability space, and what are the trade-offs?

---

> ### Author Response · Authors · 2024-11-29
>
> We want to thank the reviewer for the thoughtful review and for providing the opportunity to improve the submission with this feedback. We have worked hard in the past couple of weeks to address this feedback, and update our manuscript as outlined below!
>
> Regarding the weaknesses surrounding the lack of optimality with respect to the median split strategy, and the convergence bounds for imperfect PDF estimation, we agree that the lack of theoretic analysis is a bit strange for a paper on search algorithms. The initial reasoning for not including this analysis was that we present a general framework in which search space density estimation can be performed, and thus we did not want to make assumptions on distributions and accuracy when conducting this analysis as such analysis would heavily rely on those assumptions. To address this weakness however, we have now added a theoretical analysis section to the appendix in which we provide a convergence proof for the generalized bisection of density space with a tolerance. Additionally, we have added a convergence proof for the worst case convergence of bisecting density space on a perfectly approximated normal distribution.
>
> We expect the theoretical analysis of BBS to be quite an interesting topic for research in the coming years and are looking forward to learning more/collaborating with people that may be interested in that in the future.
>
> In our revised manuscript, we have added both Beta and Lognormal distributions to the appendix, which show consistent benefits to bisection in probability density space for additional real world distribution patterns.
>
> In our revised manuscript, we have added the results for using GPR in addition to our RF/KDE estimation. The GPR here significantly underperforms the RF/KDE in this scenario, and is consistent with previous documentation about real world limitations of using GPRs.
>
> We found this application on the Lightning Network by chance in our work, however other problems that would benefit most from BBS are bisection/search problems in which there is available data to learn the PDF, and there is a tangible real world benefit to shaving some percentage points in efficiency on the search, due to the search space being expensive, domain constrained, or even risky to access in some way. Some interesting candidates include those that are in the bisection literature, including medicine/computational biology (estimating median lethal dose, etc.), amongst others mentioned here. We have not considered other splitting strategies beyond the median splitting, but think that would be a fascinating area for exploration in the future.

---

### Official Review · Reviewer_vb7P · 2024-11-01

**Soundness:** 2
**Presentation:** 1
**Contribution:** 2
**Rating:** 3
**Confidence:** 5

**Summary:**

The authors present a modified version of the standard binary search algorithm by incorporating probabilistic techniques. They consider the input to their Bayesian Binary Search (BBS) algorithm as an ordered sequence of integers, $x_1, \ldots, x_n$, where each $x_i$ is assumed to be an IID sample from an unknown distribution $\mathbb{P}$.

If the distribution $\mathbb{P}$ were known and had a density function $p$, binary search could be optimized by using the median $m$ of the distribution as the pivot:
$$
\int_{-\infty}^m p(x) \, \mathrm{d}x = 0.5
$$
instead of using the midpoint of the current upper and lower bounds in the sequence.

Since the distribution $\mathbb{P}$ is typically unknown, the authors propose estimating the density of the sequence $x_1, \ldots, x_n$ to obtain an approximation $\hat{\mathbb{P}}$ of $\mathbb{P}$, using it as a surrogate in their probabilistic binary search algorithm.

The authors conduct an empirical evaluation of the algorithm, using synthetic data generated from a discretized univariate Gaussian distribution and a real-world dataset from the Lightning Network.

**Strengths:**

BBS is shown to improve efficiency in non-uniform search spaces, as evidenced by lower average steps to reach target values compared to standard binary search, especially in scenarios with a known or estimable distribution of search targets.

The real-world example on the Lightning Network demonstrates a potential real-world use of the algorithm, if implemented appropriately.

**Weaknesses:**

There is nothing "Bayesian" about the proposed algorithm. Perhaps a more appropriate name would be "Probabilistic Binary Search".

On the Lightning network experiment:

The primary weakness of the experiment lies in a misalignment between the stated motivation for Bayesian Binary Search (BBS) and its actual implementation.

- The experiment does not apply BBS to actual probing in the Lightning Network, where probing costs (in terms of network resources and latency) are the main concern. Instead, they simulate probing by using inexpensive predictions from a random forest model, which removes the practical need for BBS to reduce costly network probes.

- Since predictions from the random forest are computationally cheap, the added complexity of BBS (which relies on density estimation and probabilistic bisection) offers limited benefit in this context.

- There is no reported raw computation time. The added computational overhead of first performing density estimation would likely be much larger than the marginal gains achieved when performing BBS over binary search.

In summary, the experiment does not convincingly demonstrate BBS’s utility in addressing real-world probing expenses. Without actual network interaction, it fails to showcase how BBS would reduce probing costs in a live setting, which is the central argument for its use.

**Questions:**

Have the authors considered demonstrating the use of BBS in a live, online setting, where probing costs are relevant? Since the current experiment is based on offline predictions, it’s unclear if the BBS approach would indeed reduce costs in a real-world probing scenario within the Lightning Network.

---

> ### Author Response · Authors · 2024-11-29
>
> We want to thank the reviewer for the thoughtful review and for providing the opportunity to improve the submission with this feedback. We have worked hard in the past couple of weeks to address this feedback, and update our manuscript as outlined below!
>
> We want to clarify our real world experiments to address the reviewer’s primary weakness. We have a dataset collected throughout the entire network  (self reported node operator data) containing individual channel balances at a given point in time. We make the predictions (testing set) using a model which amounts to supervised density estimation. We then are able to bisect using the PDFs provided by the channel balance interpolation model. Given the setup of this experiment, the number of steps required in the search exactly matches the real world scenario. We only operate a single node on the network, so we were unable to fully reach every single channel in our dataset, which is why we structured the experiment this way.
>
> To further provide some more insight here however, we have added real probe times for 1500 channels reachable from our nodes (average of 3.1 seconds per probe per channel) to the revised manuscript and contextualized this with density estimation time (0.18 seconds on average per channel).
>
> For this use case however, the benefit goes beyond even just computation time. Each probe in our situations occupies an HTLC, which is capped per channel at 483 for the most popular Lightning node implementation, LND. This means that each probe occupies part of this cap that could otherwise be used to make a real payment, fundamentally constraining the throughput of the Lightning Network as a whole. In addition to time, and occupying HTLCs, probes consume network wide resources such as compute, bandwidth, and memory.
>
> In our use case in the Lightning Network, we are not all constrained by the time to compute this search space density as this can be done once (single model for the entire network), but are constrained during the actual search process itself, by time, liquidity, and network throughput (max htlc limit). In fact we could precompute the binary search tree beforehand, and just traverse it during the search process.
>
> As experienced Lightning node operators we feel that we are confident in our belief that this algorithm has real world impact for the reasons above, and our direction in writing this paper was because we were motivated to tackle this problem in the industrial context.
>
> We took inspiration from Bayesian Optimization, in which a surrogate function is estimated using probabilistic machine learning, when thinking about naming this general framework, and thought about the similarities in that regard with Bayesopt. Many of the methods for density estimation have Bayesian components (GPR, BNN, Bayesian parameter estimation etc), and the extension of the framework for adaptive PDF estimation can also be Bayesian updates depending on the surrogate function.

---

### Official Review · Reviewer_4wAb · 2024-11-04

**Soundness:** 1
**Presentation:** 1
**Contribution:** 1
**Rating:** 1
**Confidence:** 3

**Summary:**

This work proposes to use bayesian method to do binary search where the midpoint is not equal to the average of an internal, in contrast to the classical binary search method.

Although this idea sounds interesting, I am not convinced by the proposed method, either based on the experimental results or the presentation.

Experiment:
The authors spent so much time listing all variants of binary search in the introduction. However, during the experiments, the authors only compare with the most basic baseline, ignoring everything the authors have discussed at the beginning.

**Strengths:**

The authors ask a good research question --- how we should perform binary search when the uniform distribution assumption does not hold.

**Weaknesses:**

Lack of baselines. The authors discuss many variants of binary search during introduction. However, none of them are used as baselines for comparison other than the classical binary search method.

Plus, the proposed method described here is not very different from doing active learning using Gaussian Process.

One more thing, doing density estimation can be time consuming. This could ultimately slow down the the search process and be much slower than the classical binary search method.

**Questions:**

Can you compare with all baselines you listed in your introduction paragraph and report results during rebuttal?

How is your method different from active learning via the help of Gaussian Process?

---

> ### Author Response · Authors · 2024-11-29
>
> We want to thank the reviewer for the thoughtful review and for providing the opportunity to improve the submission with this feedback. We have worked hard in the past couple of weeks to address this feedback, and update our manuscript as outlined below!
>
> The probabilistic bisection literature mentioned (Waeber, Jedynak) is for noisy response problems (noisy oracle, so this is not the same problem we are tackling where we assume we are able to get noise free observations from the oracle, and thus we believe benchmarking to these algorithms is inappropriate.
>
> The binary search research such as interpolation, is defined on a sorted array, for which the target is known up front, rather than for bisection in which we do not know the value up front, but rather have to iteratively narrow the search space till the desired interval is obtained. We derived these experiments working backwards from our real world problem (Lightning Network), and thus the method and experiments more closely reflect mathematical bisection/root finding problems which most computer scientists still understand to be a binary search, rather than the traditional binary search on a sorted finite array although the two are very closely connected in concept.
>
> Neither the existing binary search on sorted array literature or bisection literature provided methods to incorporate our probabilistic understanding of the search space into the search process, making us believe this method was best suited for this kind of problem. As a result, the industrial approach here is typically standard bisection, as we believe is still very common in industrial applications in which assumptions can’t be made on distribution.
>
> One way to contextualize this benchmarking is to think about what methods would be used in the traditional “Number Guessing Game” where the player has to guess a secret number and the oracle can only tell the player if the guessed number is less than or equal to the secret number or greater than. From our literature survey, standard bisection is the default optimal method for this sort of problem and no other literature in our survey has incorporated methods estimating search space density which subsequently optimize the search process.
>
> We believe that active learning via the help of a Gaussian Process would fall under the general BBS framework. This would constitute supervised density estimation via the GPR, but search would happen here via finding the new median after each PDF update rather than bisecting in density space. Our framework here provides more flexibility for other methods in density estimation including those without supervision.
>
> In our use case in the real world, we are not constrained by the time to compute this search space density, but are constrained during the actual search process itself, by time, liquidity, and network throughput (max htlc limit). In fact we could precompute the binary search tree beforehand, and just traverse it during the search process.  o further provide some more insight here however, we have added real probe times for 1500 channels reachable from our nodes (average of 3.1 seconds per probe per channel). The density estimation per channel is only 0.18 seconds on average.

---

### Author Response · Authors · 2024-11-29
**Clarifying Context/Contribution**

We want to thank the reviewers for the thoughtful review and for providing the opportunity to improve the submission with this feedback. The authors would first like to provide some helpful context that we initially encountered this search problem in a real life context (Lightning Network), in which there was a real bisection problem for which we had previously developed a supervised machine learning model to predict the search target (Lightning channel balance). Each search space access here had a time cost, capital cost, and a fundamental domain scalability cost (max htlc limit for Lightning Nodes [akin to max incoming connections for Linux servers], fundamentally impacting network wide payment throughput). Such probing happens at large scale today (millions per day and growing with the network) by large industrial players and startups alike.

[Max HTLC Limit:] (https://docs.lightning.engineering/the-lightning-network/multihop-payments/hash-time-lock-contract-htlc)

The current industrial procedure is to use standard bisection, so we were inspired to understand how we could somehow leverage this ML model to improve the efficiency of search. Ultimately we realized that converting the model into a supervised probabilistic machine learning model which outputs a distribution would allow us to bisect in probability density space, allowing us to leverage our model’s knowledge about this distribution to guide the search.

This prompted us to delve into the literature to see if this was written about, and we were unable to find work that combined this sort of statistical learning with binary search in this way prompting us to write this paper and submit to this conference.

We started by trying to find out what gains would be achieved by bisecting in probability density space vs. regular bisection on a normal distribution, and to our surprise no closed form solution for the convergence of bisecting density space, nor any numerical/experimental results were found in the literature. Even without supervised/unsupervised density estimation, we could not find bisection in the density space well defined in the literature, which made sense given it only makes sense to use if there is a reason to believe a search space is distributed a certain way.

In contextualizing this contribution we primarily believe this to be a binary search/bisection paper via applied machine learning/statistical learning rather than an ML theory paper itself. The way this problem was presented to us in real life was more akin to a number guessing game in a bounded continuous space, in which the oracle would be able to tell us if our search target was above or below the probe, but not when we exactly reached the probe, allowing us to search till we were confident the target was within a given range (epsilon). As such the experiments are structured in this way, impacting our benchmarking approach. We hope this explanation clarifies the contribution of the paper and our general approach.

---

### Meta-Review · Area_Chair_bhj2 · 2024-12-29

**Metareview:**

This paper presents Bayesian Binary Search (BBS), which modifies traditional binary search by using probability density estimates to guide the search process rather than midpoint splitting. It includes an interesting application to the Bitcoin Lightning Network. However, the approach received mixed reviews from strong reject (4wAb) to "marginally above acceptance threshold" (XdTs), with the average score being well below the “accept” rating..

In terms of strengths, the novel combination of probabilistic methods with binary search that showed improved efficiency in non-uniform search spaces (XdTs and vb7P for ) and the good performance on the “real-world” Lightning Network application (though vb7P raised concerns about experimental methodology).

Multiple reviewers identified consistent critical weaknesses: Reviewer XdTs and Fj8u highlighted a lack of theoretical analysis regarding optimality and convergence bounds; Reviewer 4wAb criticized the limited empirical evaluation and missing baseline comparisons.  The authors pushed back on this, but Fj8u suggested a way of adapting baselines used in the noisy context.  Reviewer 4wAb, Fj8u, (as well as myself) noted the paper's poor fit for ICLR due to its lack of connection to representation learning. Reviewer vb7P questioned computational overhead considerations.

The reviewers and the AC recommend another venue such as AISTATS or ICML where this work might be a better fit. There could be a nice paper here if the authors focused on (1) Developing theoretical foundations, specifically addressing the convergence and optimality concerns raised by Reviewers XdTs and Fj8u; (2) Expanding empirical evaluation to include the sophisticated baselines mentioned by Reviewer 4wAb and the complex distributions requested by Reviewer XdTs; and (3) Addressing Reviewer vb7P's concerns about computational overhead and terminology. The authors' responses show awareness of these issues, particularly in adding Beta and Lognormal distribution results, but more substantial development is needed in both theoretical and empirical aspects before publication.

**Additional Comments On Reviewer Discussion:**

- Reviewers (XdTs, Fj8u) criticized lack of theoretical analysis. The authors subsequently added convergence proofs for a few cases. Fj8u acknowledged improvement but wanted more comparative analysis.

- Reviewers (4wAb, XdTs) requested more distributions and baselines. The authors added Beta/Lognormal distributions and GPR comparisons

While authors made substantial improvements in theory and empirical evaluation, reviewers still felt key weaknesses remained in baseline comparisons, theoretical foundations, and presentation.

---

### Decision · Program_Chairs · 2025-01-22

Reject